# Role of First Trimester Screening Biochemical Markers to Predict Hypertensive Pregnancy Disorders and SGA Neonates—A Narrative Review

**DOI:** 10.3390/healthcare11172454

**Published:** 2023-09-01

**Authors:** Wojciech Górczewski, Joanna Górecka, Magdalena Massalska-Wolska, Magdalena Staśkiewicz, Dariusz Borowski, Hubert Huras, Magda Rybak-Krzyszkowska

**Affiliations:** 1Independent Public Health Care Facility “Bl. Marta Wiecka County Hospital”, 32-700 Bochnia, Poland; 2Department of Obstetrics and Perinatology, University Hospital, 31-501 Krakow, Poland; 3Clinical Department of Gynecological Endocrinology and Gynecology, University Hospital, 31-501 Krakow, Poland; 4Clinic of Obstetrics and Gynecology, Provincial Combined Hospital in Kielce, 25-736 Kielce, Poland; 5Department of Obstetrics and Perinatology, Jagiellonian University Medical College, 31-501 Krakow, Poland

**Keywords:** high-risk pregnancies, biochemical markers, hypertensive disorders in pregnancy, small-for-gestational-age, PlGF, PAPP-A, narrative review

## Abstract

Early recognition of high-risk pregnancies through biochemical markers may promote antenatal surveillance, resulting in improved pregnancy outcomes. The goal of this study is to evaluate the possibilities of using biochemical markers during the first trimester of pregnancy in the prediction of hypertensive pregnancy disorders (HPD) and the delivery of small-for-gestational-age (SGA) neonates. A comprehensive search was conducted on key databases, including PubMed, Scopus, and Web of Science, for articles relating to the use of biochemical markers in the prediction of HPD and SGA. The findings show that changes in the levels of biomarkers in the early pregnancy phases could be an important indicator of adverse pregnancy outcomes. The literature shows that low PAPP-A (pregnancy-associated plasma protein A) and PlGF (placental growth factor) levels, low alkaline phosphatase (AP), higher sFlt-1 (soluble fms-like Tyrosine Kinase-1) levels, higher AFP (alfa fetoprotein) levels, and elevated levels of inflammatory markers such as β-HGC (free beta human chorionic gonadotropin), interferon-gamma (INF-γ), and tumor necrosis factor-α (TNF-α) may be associated with risks including the onset of HPD, fetal growth restriction (FGR), and delivery of SGA neonates. Comparatively, PAPP-A and PlGF appear to be the most important biochemical markers for the prediction of SGA and HPD.

## 1. Introduction

Hypertensive disorders in pregnancy (HPD) and abnormal patterns of growth of the fetus, such as small-for-gestational-age (SGA), are among the leading causes of perinatal and maternal morbidity and mortality in the world. These conditions often contribute to long-term negative effects on the mother and the unborn child. Globally, gestational hypertension affects between 5% and 8% of all women [1]. However, pregnancy-induced hypertension may occur at a frequency of up to 16.7% in developing countries [2]. In Poland, Lewandowska and Więckowska have reported a prevalence of up to 12.4% of pregnancy-induced hypertension [3]. In many of these cases, it causes poor perinatal outcomes as well as a higher risk of complications during pregnancy such as high blood pressure, eclampsia, preeclampsia (PE), placental abruption, and stroke [4,5]. In addition, HPD may cause neonatal and maternal deaths [4]. It has also been associated with elevated risks of fetal asphyxia, low birth weights, preterm delivery, and stillbirth [2]. Therefore, HPD is one of the major problems in perinatal medicine and public health.

The occurrence of small-for-gestational-age (SGA) is frequently associated with HPD. The classic paradigm placing the placenta in the role of the villain is based on the statement that the abnormally progressive physiological process of vascular remodeling leads to endothelial dysfunction and consequently to the development of hypertension during pregnancy with all its subtypes and consequences. However, coexisting abnormal maternal cardiovascular function before pregnancy may be a contributing factor to the development of HPD, especially preeclampsia, putting the malperfused trophoblast in the role of victim [6,7,8,9]. One of the causes of SGA is a decrease in placental blood perfusion. During pregnancies complicated by HPD, high resistance flow through the spiral arteries results in impaired placental perfusion, which contributes to local ischemia and an increase in the malnutrition rate of fetuses with varying degrees of expressed intrauterine growth disorders [10].

Placental ischemia appears to stimulate the release of antiangiogenic factors such as sFlt-1, soluble endoglin (sEng), and endothelin 1 (ET-1), subsequently increasing their plasma levels relative to proangiogenic factors, namely vascular endothelial growth factor (VEGF) and PlGF. In addition, placental ischemia enhances the concentration of pro-inflammatory cytokines such as tumor necrosis factor (TNF), interleukin-6 (IL-6), hypoxia-inducible factor (HIF), reactive oxygen species (ROS), and agonistic autoantibodies (AT 1AA). When these bioactive factors are secreted locally near the placenta, they may target uterine and placental matrix metalloproteinases (MMPs), further adversely affecting placental development and vascularization. This may exacerbate placental ischemia, potentially leading to fetal growth anomalies [11].

Conversely, when these bioactive factors enter the maternal circulation, they can target the endothelium, vascular smooth muscle (VSM), vascular MMPs, and extracellular matrix (ECM). This can result in generalized vasoconstriction, endothelial dysfunction, and the development of a systemic maternal response. Hypertension may affect the integrity of the endothelium in the uterine vessels of the pregnant mother and subsequently result in fetal growth restriction [12]. Other hypertension-related complications, such as intravascular coagulation and histopathologic lesions (mostly massive perivillous fibrin deposition), may also affect blood flow to the fetus and therefore affect normal growth. Furthermore, elevated blood pressure may hamper the development of the placental villous tree [10]. This results in a general decline in placental function and a decrease in fetal growth. Therefore, gestational hypertension is a major risk factor for the birth of SGA neonates.

Hypertensive women are a high-risk pregnancy group who need close clinical monitoring. As such, early identification of the occurrence of gestational hypertension and SGA is critical for the monitoring of this group to facilitate the initiation of preventive measures. The traditional approach to evaluating the risk of developing gestational hypertension in pregnant women involves the analysis of maternal demographic features and clinical presentations to predict risk factors [13]. This screening model focuses on assessing the risk of gestational hypertensive disorders during the first trimester of pregnancy by understanding maternal characteristics such as age and body mass index (BMI), as well as available obstetric and medical history [14]. While this approach remains in use in clinical settings, it may not be an accurate method of predicting gestational hypertension and SGA. Luckily, more sensitive, high-accuracy screening methods using biochemical markers have emerged that may predict preeclampsia, a subtype of HPD in pregnancy. Such screening is necessary for the management and prevention of risks associated with HPD. For instance, the administration of a daily dose of aspirin not later than 16 weeks until 36 weeks has been shown to reduce the incidence of gestational hypertension and preeclampsia in pregnancies with a high risk of developing preeclampsia based on predictive algorithms [15]. In the context of screening for preeclampsia in the first trimester, the standard practice of administering aspirin is influenced by the data related to arterial pressure and blood flow in both uterine arteries, which is consistent with the findings from the ASPRE trial [16]. Modern predictive models for PE according to the Fetal Medicine Foundation’s algorithm, including history with risk factors, uterine artery flows, and blood pressure measurements, determine the risk of PE and FGR, and the inclusion of biochemical markers such as PAPP-A or PlGF in this risk calculation further increases the sensitivity of the method. A risk ratio above 1:100, according to the ASPRE trial, is an indication for low-dose aspirin. The recommended dose and timing of use vary depending on the recommending scientific society; for example, SOMANZ, ISH, and ESC recommend starting aspirin at 12 weeks but definitely before 16 weeks, while ending aspirin use between 36 and 37 weeks of pregnancy, while NICE, SOCG, and ACOG recommend aspirin until delivery. Additionally, the optimal aspirin dosage is inconsistent. NICE recommends 75–150 mg/d; ISH, ISSHP, and SOCG recommend 75–162 mg/d; ESC and PTGiP recommend 100–150 mg/d; FIGO and ISUOG recommend 150 mg/d; ACOG and USPSTF recommend 81 mg/d [17,18,19,20,21,22,23,24,25,26], while WHO recommends 75 mg/d. Data from a systematic review and metanalysis indicate that doses below 100 mg/d do not improve obstetric outcomes [27], and 10–30% of patients are resistant to aspirin at doses up to 81%, according to a prospective study by Caron et al. [28]. Therefore, first-trimester screening for gestational and other forms of hypertensive disorders during pregnancy increases the risk of prematurity for a neonate and the higher risk of cardiovascular complications in mothers many years after they suffer from PE.

Biochemical markers have a strong potential for use in predicting gestational hypertension and, subsequently, the risk of SGA births. They provide a convenient, non-invasive method of screening for aneuploidies, PE, and FGR [29]. During gestation, the placenta of a pregnant woman releases various compounds and factors into maternal blood circulation, like steroid and peptide hormones, which promote angiogenesis. These factors play a critical role in regulating maternal metabolism during pregnancy. They also promote physiological adaptations for maintaining pregnancy and ensuring successful fetal growth and development [30]. These factors that are present in maternal blood may serve as biochemical indicators. They can also be used for the early prediction of pregnancy disorders. This is due to the fact that the serum concentrations of gestational compounds in maternal blood may increase or decrease in response to the onset of certain physiological or pathological conditions. For instance, Poon and Nicolaides estimate that first-trimester PAPP-A, PlGF, and other factors including maternal history, MAP (mean arterial pressure), and uterine artery Pulsative Index (PI) may identify up to 95% of cases of early onset of preeclampsia [31]. Thus, biochemical markers in the early phases of pregnancy may be used to predict the occurrence of gestational hypertension. Most of these complications occur during the later stages of gestation but are caused by physiological and pathological mechanisms that occur in the first weeks of pregnancy. As a result, early recognition of high-risk pregnancies through the detection of biochemical markers may promote antenatal surveillance, resulting in an overall improvement in pregnancy outcomes. Hence, the goal of this study is to evaluate the possibilities of using biochemical markers identified during the first trimester of pregnancy in the prediction of the occurrence of gestational hypertension and the delivery of SGA neonates.

This narrative review aims to explore and evaluate the utility of biochemical markers typically used in preeclampsia screening, assessing their potential applicability and effectiveness in the detection of gestational hypertension and SGA neonates. Our objective is to establish an evidence-based correlation between these markers and the incidence of gestational hypertension, potentially offering a novel approach for early detection and better management of this common pregnancy complication. In particular, we focus on the most clinically adapted biochemical markers that have entered widely-used practice.

## 2. Materials and Methods

### 2.1. Study Design

A comprehensive search was conducted on key medical and health databases, including PubMed, Scopus, and Web of Science, for articles relating to the theme of biochemical markers in the prediction of gestational hypertension and SGA births. Several keywords and their combinations were used to retrieve articles from these databases, including gestational hypertension, small-for-gestation-age, SGA, biochemical markers, biomarkers, biological markers, and first trimester.

### 2.2. Selection Process

Eligible studies were those that included the quantification of biochemical markers in the serum of pregnant participants. The outcomes of interest in this study were the means of blood pressure of pregnant cohorts as well as the weights, lengths, and head circumferences of neonates. Studies were excluded from the empirical review if they included qualitative assessments. Furthermore, publications that were not in English as well as studies involving animals were excluded from the empirical review. Using these search criteria, a total of 15 primary studies were identified that formed the basis of this empirical review (Figure 1).

### 2.3. Variables and Measurement

Gestational hypertension: Hypertension is characterized by a systolic blood pressure of 140 mm Hg or above and/or a diastolic pressure that is equivalent to or exceeding 90 mm Hg [21]. Various types of HPD occur during pregnancy in which the blood pressure is abnormally elevated. One of these types is gestational hypertension, which refers to a hypertensive disorder of pregnancy that occurs after a gestation period of 20 weeks and in the absence of proteinuria or abnormalities of a biochemical or hematological nature [21]. Gestational hypertension is often not accompanied by restrictions on fetal growth. The condition resolves within 42 days after childbirth [32]. Pregnancy-related hypertension may also present in the form of preeclampsia, which is defined as the onset of hypertension after the 20th week of gestation that is accompanied by one or more end-organ involvements, such as renal liver dysfunction, hematologic complications, neurologic complications, uteroplacental dysfunction, and abnormal laboratory results including liver enzymes, platelet count, DIC, and hemolysis [17,18,19,20,21,22,23].

SGA refers to neonates who are small-for-their-gestational age. It describes neonates whose length, head circumference, and weight are lower than normal standards for their age [33]. For the purposes of this study, the SGA criterion is established under the standards of the INTERGROWTH-21st Project [33,34]. With respect to birth length, SGA newborns are defined as those below the third centile [34]. The INTERGROWTH-21st criteria further identify birth weights below the 10th percentile as constituting SGA. A severe SGA may occur if the abdominal circumference (AC) or estimated fetal weight (EFW) is below the third percentile [35]. Lees et al. distinguish between SGA and fetal growth restriction (FGR) [36]. They adopt the Society for Maternal–Fetal Medicine’s (SMFM) criteria of viewing the SGA as a neonate whose abdominal circumference or estimated fetal weight is lower than the 10th percentile of the normal range, while the FGR is a fetus that does not attain its genetically predetermined growth capacity [36]. According to the International Society of Ultrasound in Obstetrics and Gynecology (ISUOG), a fetus is considered to have FGR if it has AC and EFW measurements below the 10th percentile, along with the existence of abnormal Doppler outcomes [37].

Biochemical markers: These are factors, compounds, or molecules released during physiological gestation into the maternal circulation whose role is to aid in the regulation of maternal metabolism during pregnancy [30]. They include beta-human chorionic gonadotropin, pregnancy-associated plasma protein A, placental growth factor, and alkaline phosphatases.

## 3. Results

Honarjoo et al. examined the role of pregnancy-related biomarkers in predicting SGA births [38]. They focused on two biomarkers, namely β-hCG and PAPP-A. A cohort of 16 participants selected randomly from health centers in Iran was examined. Screening test data for first-trimester fetal anomalies were obtained for the cohort and analyzed. Specifically, data on the SGA odds ratio and levels of biomarkers in the serum were compared to determine whether the two variables had any correlation. The study findings revealed that low PAPP-A levels were associated with a statistically significant increase in the risk of developing SGA. However, β -hCG levels did not have a significant correlation with SGA risk [38].

Allen and Aquilina conducted a prospective study to assess the accuracy of serum biomarkers released in the first semester in the prediction of adverse pregnancy outcomes, such as gestational hypertension and delivery of small-for-gestational-age neonates. They obtained a large sample of 1045 women who attended first-trimester scans at the Royal London Hospital. Participants’ maternal histories, such as weight, height, age, ethnicity, past medical histories, and drug usage, were collected. Maternal blood samples were also taken to test for biomarkers including PAPP-A, β-hCG, PlGF, and Alpha-fetoprotein (AFP). Pregnancy outcomes such as preeclampsia, gestational hypertension, stillbirth, and SGA were assessed for the sample for multivariate analyses to determine the correlation between the variables. The findings revealed that low levels of PlGF and PAPP-A were significantly related to the risk of SGA. AFP and β-hCG biomarkers were not found to be significantly associated with SGA or gestational hypertension. While the study showed a potentially beneficial role for biochemical markers in predicting SGA, it was limited by the low prevalence of outcomes, with preeclampsia and hypertension having incidences of 1.3% and 2.2%, respectively [39].

In another study, Hendrix et al. examined whether first-trimester maternal biomarkers could improve the prediction of SGA and other adverse neonatal outcomes [40]. Their research involved a retrospective cohort study in which singleton pregnancies of patients attending routine check-ups at the Maastricht University Medical Centre, Maastricht, the Netherlands, were examined. All singleton pregnancies were included for the period from 2012 to 2016, resulting in a final sample of 296 participants. They measured four biomarkers, namely β-hCG, PAPP-A, PlGF, and sFlt-1, between 11 and 13 weeks of gestational age. In addition, fetal growth scans were performed towards the end of the gestational period, and the birth weight percentile was calculated for SGA determination. The serum biomarkers were correlated with fetal growth outcomes to determine their screening abilities. The findings revealed that a significantly higher sFlt-1 was associated with the risk of delivering SGA neonates. The risk of SGA delivery was also found to increase for mothers whose blood serum had higher sFlt-1/PlGF MoM ratios. Lower PAPP-A values were also associated with the risk of SGA. The study demonstrated the role of biochemical markers in predicting the risk of SGA. However, the study was limited by its relatively small sample size [40].

Papastefanou et al. have also used first-trimester biochemical markers to predict small-for-gestational-age neonates [41]. They combined the biochemical markers with maternal and biophysical factors. The study involved a large prospective investigation with 60,875 participants. All participants were women with singleton pregnancies attending routine examinations in the first trimester at two UK hospitals, namely the Gillingham-based Medway Maritime Hospital and the London-based King’s College Hospital. Data on serum biomarkers of PAPP-A and PlGF were obtained for all participants. Pregnancy outcomes such as birth weight percentiles at delivery were obtained from hospital records. The data was used to develop likelihood functions for predicting pregnancy outcomes based on biochemical markers. The findings revealed that placental growth factor (PlGF) was the best biochemical marker for the prediction of SGA. Additionally, PlGF and PAPP-A were found to predict 48.6% of all SGA neonates with preeclampsia delivered before the age of 32 weeks. These markers also predicted 59.1% of the risk of SGA neonates with preeclampsia at 32 weeks of gestation. In the case of SGA without preeclampsia, biochemical markers predicted 44.4% and 51.2% of SGA neonates for deliveries at 37 weeks and 32 weeks, respectively. The study had the strength of having one of the largest sample sizes [41].

PAPP-A tests have also been widely used in predicting FGR and SGA. Multiple studies have shown that low PAPP-A levels are associated with adverse outcomes in pregnancy. For instance, Boutin et al. examined the correlation between PAPP-A levels and the incidence of adverse perinatal outcomes. They obtained a sample of nulliparous women from a low-risk population comprised of singleton pregnancies. A large sample of 4739 participants was recruited and examined at a university hospital in Quebec, Canada. Serum samples from participants were obtained at 11–13 weeks and measured for PAPP-A levels in multiples of the median (MoM). Follow-up was made for all participants for pregnancy outcomes such as SGA, preeclampsia, and fetal fatality. The results showed a moderate association between PAPP-A levels and the incidence of preeclampsia and SGA. However, PAPP-A levels were not correlated with neonatal death. PAPP-A values below 0.4 MoM raised the risk of SGA and preeclampsia substantially, though the predictive value was low. The authors concluded that PAPP-A is used in combination with other biochemical markers in preeclampsia and SGA prediction [42].

Belovic et al. examined the role of placental biochemical markers in predicting gestational hypertension and preeclampsia [30]. They estimated the capacity of various biochemical markers to predict pregnancy-induced hypertension. Baseline ultrasonographic data were obtained for a sample of pregnant women, after which regular check-ups were performed to determine the changes in blood pressure, BMI, and the onset of hypertension. In addition, serum biochemical markers were obtained in multiples of the median (MoM). Specifically, the study collected data on β-hCG, inhibin, alpha-fetoprotein (AFP), and PAPP-A during the follow-up period. The findings revealed a high prevalence of hypertensive disorders in 20.2% of the population. Moreover, a significant correlation was observed between first-trimester PAPP-A levels and the risk of gestational hypertension. Participants with preeclampsia were found to have significantly higher AFP levels. Thus, the biomarkers were found to be useful for predicting gestational hypertension and preeclampsia [30].

Sharma et al. investigated the role of inflammatory biomarkers released during the first trimester in predicting the risk of gestational hypertensive disorders [43]. Their study focused on the efficacy of PAPP-A, β-HCG, interferon-gamma (INF-γ), and tumor necrosis factor-α (TNF-α). A prospective study was conducted in two phases. In the first phase, a cohort of 2000 pregnant women was examined and followed up until delivery. The second phase involved a case-control study in which participants diagnosed with hypertension in the first phase were evaluated for pro-inflammatory markers. The results showed that the population had a low prevalence of gestational hypertension (9.13%) and preeclampsia (2.72%). Moreover, there was a significant correlation between low PAPP-A serum levels and gestational hypertension. Furthermore, high β-hCG levels were associated with a risk of gestational hypertensive disorders, though the relationship was not statistically significant. Similarly, high TNF-α and INF-γ levels were observed in individuals with gestational diabetes. A combination of all the biomarkers had high predictive values of 71.4% for positive results and 51.6% for negative outcomes [43].

Extreme values of serum biomarkers in the first trimester have been associated with elevated risks of adverse pregnancy outcomes. For instance, Genc et al. investigated the relationship between extreme values of maternal biomarkers in the first trimester and adverse pregnancy outcomes. The study involved an analysis of 786 pregnant women who attended routine perinatal screening at a hospital in Turkey. They specifically analyzed two biochemical markers, namely PAPP-A and β-hCG. The findings of the study revealed that there was a statistically significant correlation between PAPP-A and the presence of SGA neonates. Low serum β-hCG was also found to increase the risks of adverse obstetric outcomes. Thus, the two biomarkers were identified as useful biomarkers for predicting the risks of fetal growth restrictions [44].

Wang et al. examined the association between oxidative stress, serum β-hGC, and pregnancy-induced hypertension [45]. Their study involved an examination of a cohort of pregnant women. The sample included non-hypertensive individuals as well as patients with various severities of preeclampsia. Upon follow-up, the authors demonstrated that high levels of serum β-hCG were correlated with the severity of pregnancy-induced hypertension. Similarly, the risk of pregnancy-induced hypertension was higher in individuals whose blood serum had elevated levels of oxidative stress factors such as TNF-α, interleukin-6, and interferon-γ. Thus, the authors concluded that the presence of β-hCG levels alongside oxidative and inflammatory markers could be used to predict pregnancy-induced hypertension [45]. These findings are similar to those of Elazab et al., who recently demonstrated that a combination of TNF-α, β-hCG, and lipid profiles were useful biomarkers for predicting the severity of preeclampsia [46]. Their research involved a case-controlled study in which 90 women were recruited for assessment of the role of biochemical markers in the prediction of preeclampsia. The study findings revealed that there was a statistically positive correlation between the serum levels of triglycerides, cholesterol, low-density lipoprotein, β-hCG, and TNF-α and the severity of preeclampsia. Thus, these biomarkers could be used to predict the risk of the onset of preeclampsia [46].

Some studies have shown that biochemical markers may be less effective in predicting FGR and gestational hypertension. For instance, Khanam, Mittal, and Suri evaluated whether the addition of β-hCG and PAPP-A biomarkers to the Uterine Artery Pulsatility Index (UtA-PI) in the first trimester could be used in predicting preeclampsia. They conducted a prospective observational study in which 100 pregnant women were recruited. For these participants, PAPP-A, β-hCG, and UtA-PI levels were measured at the gestational age between 11 and 14 weeks. A follow-up was made for all participants to assess the incidence of gestational hypertension and preeclampsia. The findings revealed that UtA-PI was an effective biomarker for screening hypertension and preeclampsia and that the inclusion of PAPP-A and β-hCG did not enhance the predictive value for these conditions. As such, the addition of PAPP-A and β-hCG to UtA-PI was not recommended for routine screening [47]. However, the study had the limitation of having a relatively small sample. In another study, Nicolaides et al. demonstrated that the peak systolic velocity ratio was superior to maternal biochemical factors in predicting preeclampsia [48].

Zumaeta et al. compared the efficacy of PAPP-A with placental growth factor in screening for preeclampsia in the first trimester [49]. They also evaluated the screening abilities of both biomarkers. Their investigation involved a non-intervention screening study in which data was obtained from patients attending routine visits at two UK hospitals. Maternal characteristics and medical histories, as well as MoM values of mean arterial pressure, PlGF, UtA-PI, and PAPP-A, were also obtained for the sample. Positive detection rates were compared between screenings involving PlGF and PAPP-A and those without these biomarkers. They examined data from 60,875 participants, of whom 1736 developed preeclampsia, accounting for 2.9% of the population. The findings showed that a screening test involving the combination of mean arterial pressure, UtA-PI, PlGF, and maternal factors was superior to those where PlGF was absent. They also established that the addition of PAPP-A biomarkers did not improve the sensitivity of the screening method. Thus, the authors concluded that PlGF was a preferred biochemical marker instead of PAPP-A in the first-trimester screening of preeclampsia [49].

Low alkaline phosphatase levels have also been associated with hypertensive disorders and are therefore used as biomarkers for predicting perinatal outcomes. For instance, Tang et al. examined various biochemical markers, including alkaline phosphatase, in the prediction of preeclampsia [50]. They conducted a retrospective cohort study in which data from 20,582 women was obtained and analyzed. The cohort included both pregnant women with preeclampsia and healthy individuals. The demographic features, blood pressure, and biochemical test data were compared with perinatal outcomes for the sampled singleton pregnancies. The findings revealed that elevated uric acid and alkaline phosphatase were associated with the risk of preeclampsia. Thus, alkaline phosphatase levels could be used to predict pregnancy outcomes [50]. In an earlier study, Duan et al. demonstrated that serum alkaline phosphatase levels were significantly associated with preeclampsia risk and severity [51]. Duan et al. conducted a retrospective case-control study in which a sample of 256 pregnancies was assessed. The participants were categorized into mild preeclampsia, severe preeclampsia, and healthy non-hypertensive individuals. Alkaline phosphatase and other serum biomarkers were measured for each of the participants in the three groups. The study findings showed significantly higher serum levels of alkaline phosphatase, lactic dehydrogenase, and D-dimer in patients with mild or severe preeclampsia compared with healthy participants. As such, these biomarkers could be used to predict the severity of preeclampsia [51].

In clinical practice, serum uric acid is widely utilized as a basic marker suggestive of various conditions, predominantly renal-related. Its correlation with HDP has been long recognized, with both European and American guidelines recommending uric acid level testing in HDP [32,52]. The ratio of serum uric acid to creatinine (SUA/sCR) may serve as a distinguishing factor for hyperuricemic patient subtypes and as an early indication of maladaptive hemodynamic changes during pregnancy, specifically in cases of HDP. Elevated SUA/sCR values at any point during pregnancy are associated with the onset of PE and adverse pregnancy outcomes, independent of the trimester in which these values are measured [53].

Endothelin-1 (ET-1) is a powerful vasoconstrictor that influences cell growth and proliferation through the activation of the Mitogen-Activated Protein Kinase (MAPK) pathway. Evaluating plasma endothelin levels during the first trimester of pregnancy can serve as an insightful marker for gauging the risk of PE. Elevated levels are observed in 55.5% of those at high risk for developing PE, and when coupled with MAP values in the mid-trimester, the positive predictive value escalates to 68.2% [6].

Finally, Chaparro et al. investigated the potential of angiogenic factors and placental biomarkers in predicting preeclampsia [54]. They postulated that endothelial and placental biomarkers present in the gingival crevicular fluid and saliva could be used as biomarkers of preeclampsia prediction. To investigate this assertion, a case-control study was conducted in which a sample of 30 patients admitted to a hospital in Chile was recruited. Of this sample, 10 had preeclampsia, while 20 had normal pregnancies. The levels of sFlt-1, placental alkaline phosphatase, and placental growth factor (PlGF) were measured for the participants. The findings revealed that the concentrations of sFlt-1 were significantly associated with the occurrence of preeclampsia. Similarly, there was a significant relationship between the concentration of placental alkaline phosphatase and preeclampsia. Individuals with preeclampsia had elevated levels of placental alkaline phosphatase. However, no differences were observed in PlGF levels between patients with preeclampsia and those with normal pregnancies. Generally, the findings show that sFlt-1 and alkaline phosphatase in saliva and gingival fluid may be used as biomarkers of preeclampsia but not PlGF. However, the study was limited by the small sample size [54].

## 4. Discussion

Despite many years of research, the clear pathogenesis of the development of HDP and, particularly, PE with or without concomitant FGR/SGA is still unknown. In view of the many biochemical markers that have been studied, only a few of them have found practical clinical application today, which is reflected in their inclusion in predictive models of PE and FGR risk calculation. Biochemical markers released during the early phases of pregnancy have important predictive value for the occurrence of gestational complications such as hypertension and preeclampsia. This is due to the fact that most of the complications occurring in the later phases of gestation can be traced to earlier physiological and pathological processes. However, in predicting gestational hypertension, changes in the biochemical markers in early pregnancy may have important implications for fetal growth. Thus, maternal biomarkers could be used to predict fetal growth restrictions and small-for-gestational-age outcomes. Therefore, early detection of changes in maternal biochemical markers may be used in the identification of high-risk pregnancies. As a result, early detection of biochemical markers could aid in the initiation of antenatal surveillance, which could facilitate the deployment of corrective measures and treatment interventions. While prior studies have documented the predictive value of biochemical markers in predicting adverse perinatal outcomes, relatively few studies have explored the impacts of using these markers to predict both SGA and gestational hypertension in the same study. Therefore, further studies are needed to evaluate the predictive value of biochemical markers such as β-hCG, PAPP-A, PlGF, and alkaline phosphatase identified during the first trimester of pregnancy in the prediction of the occurrence of gestational hypertension and the delivery of SGA neonates.

## 5. Conclusions

This review examined the extant literature on the use of biochemical markers in the first semester in predicting two perinatal outcomes, namely gestational hypertension and small-for-gestational-age births. The findings show that changes in the levels of biomarkers in the early phases of a pregnancy could be an important indicator of subsequent pathologic and physiologic changes. Specifically, the literature shows that low PAPP-A and PlGF levels, low alkaline phosphatase, higher sFlt-1 levels, higher AFP levels, and elevated levels of inflammatory markers such as β-hGC, interferon-gamma (INF-γ), and tumor necrosis factor-α (TNF-α) may be associated with the risks of adverse perinatal outcomes, including the onset of pregnancy-induced hypertensive disorders, fetal growth restriction, and delivery of SGA neonates. Comparatively, PAPP-A and PlGF appear to be the most clinically important biochemical markers for the prediction of SGA and gestational hypertension. However, mixed outcomes were reported in this review, with some studies showing no impact of the changes in some biomarkers on perinatal outcomes. Hence, there is a need for continued investigations to enhance the current understanding of the predictive value of maternal biomarkers released in the first trimester or to create other new multiparametric risk scales that will more accurately predict the risk of HDP and fetal growth abnormalities.

## Figures and Tables

**Figure 1 healthcare-11-02454-f001:**
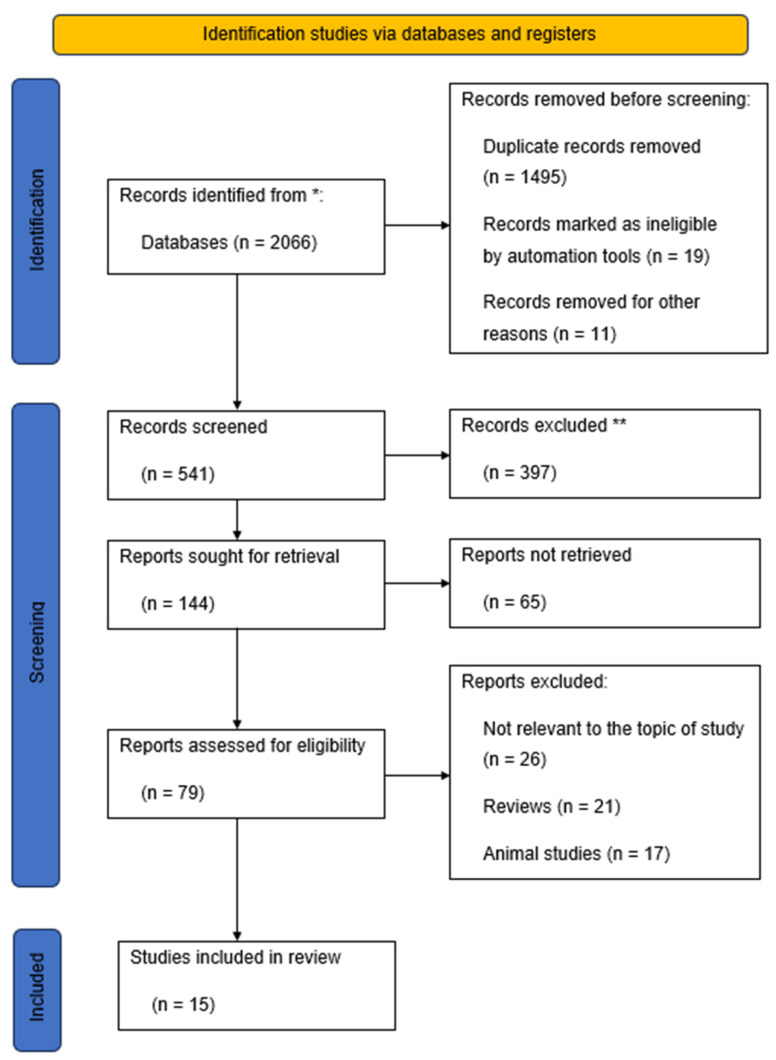
Overview of the search process. **NOTE: *** Databases: PubMed, Scopus, and Web of Science; ** Records excluded after screening titles and abstracts.

## Data Availability

Data sharing are not applicable.

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
