# Peer review of "Role of First Trimester Screening Biochemical Markers to Predict Hypertensive Pregnancy Disorders and SGA Neonates—A Narrative Review"

_healthcare, 2023, doi:10.3390/healthcare11172454_

Round 1

Reviewer 1 Report

The review by Górczewski and colleagues aims to summarise current evidence on biochemical markers for the early prediction of gestational hypertension. The topic is of great interest however the manuscript needs extensive revision since it does not accurately summarise current literature nor does provide any novel information. I have several concerns:

- The title is misleading and confusing, since it mentions biomarkers for PE prediction in the context of gestational hypertension (see below 3rd specific comment)

In the Introduction section, I would not say that hypertension precedes endothelial dysfunction. Indeed, it is more plausible that the presence of endothelial dysfunction induces the development of hypertension (doi: 10.1007/5584_2016_90). This is in line with the current hypothesis that Hypertensive disorders of pregnancy may be caused by a pre-existent endothelial dysfunction(doi: 10.1016/s0020-7292(99)00180-0doi: 10.3233/CH-221533).

- Authors defined preeclampsia with an outdated definition: “The onset of hypertension after the 20th week of gestation and the occurrence of significant proteinuria”. Proteinuria is not anymore mandatory to diagnose Preeclampsia (see latest ISSHP guidelines). This is of crucial importance because Authors should not refer to gestational hypertension alone but to overall hypertensive disorders of pregnancy (HDP). In fact, it is not possible to clearly discriminate between the two conditions giving that definitions have changed over time and frequently overlap. Indeed, in some sections of the manuscript Authors mentioned different types of HDP. I recommend Authors to modify the manuscript referring to HDP instead of gestational hypertension throughout all the text (including the title).

- Several important and widely available biomarkers studied in the first trimester for the prediction of HDP are missing (e.g. endothelin-1 and other markers of endothelial dysfunction doi: 10.1016/s0020-7292(99)00180-0., arterial stiffness doi: 10.1097/HJH.0000000000001664.). One of the most important ignored biomarker is serum uric acid and the recently evaluated serum uric acid/Creatinine ratio. Uric acid and uric acid normalized for kidney function have shown to hold prognostic value in HDP (doi: 10.1038/s41440-023-01181-6; doi: 10.1097/HJH.0000000000003472) being also included in the recommended biochemical analytes to be measured in women at high-risk (see ESC guidelines and ISSHP guidelines). 

- Authors stated that PAPP-A and PlGF are the most important biochemical markers for the first trimester prediction of HDP development. However, these well studied markers have shown in multiple high-powered studies to have poor predictive values (e.g. doi: 10.1002/pd.5001; doi: 10.1186/s12884-015-0608-y). Authors should focus on novel biomarkers and current literature on less-studied biomarkers in order to provide evidence for future studies to develop a multiparametric risk score that can ACCURATELY predict the risk for HDP development. This is of crucial importance since, as Authors stated, the only therapeutic measure is prevention and HDP incidence is dramatically increasing worldwide.

Author Response

Dear Reviewer, 

Thank you for your valuable comments. we have adjusted our work and made the suggested changes. below we respond point by point to the comments. 

The review by Górczewski and colleagues aims to summarise current evidence on biochemical markers for the early prediction of gestational hypertension. The topic is of great interest however the manuscript needs extensive revision since it does not accurately summarise current literature nor does provide any novel information. I have several concerns:

- The title is misleading and confusing, since it mentions biomarkers for PE prediction in the context of gestational hypertension (see below 3rd specific comment)

We have changed the title: Role of first trimester screening biochemical markers to predict hypertensive pregnancy disorders and SGA neonates – a narrative review

- In the Introduction section, I would not say that hypertension precedes endothelial dysfunction. Indeed, it is more plausible that the presence of endothelial dysfunction induces the development of hypertension (doi: 10.1007/5584_2016_90). This is in line with the current hypothesis that Hypertensive disorders of pregnancy may be caused by a pre-existent endothelial dysfunction(doi: 10.1016/s0020-7292(99)00180-0; doi: 10.3233/CH-221533).

We have rewritten the introduction as suggested.

- Authors defined preeclampsia with an outdated definition: “The onset of hypertension after the 20th week of gestation and the occurrence of significant proteinuria”. Proteinuria is not anymore mandatory to diagnose Preeclampsia (see latest ISSHP guidelines). This is of crucial importance because Authors should not refer to gestational hypertension alone but to overall hypertensive disorders of pregnancy (HDP). In fact, it is not possible to clearly discriminate between the two conditions giving that definitions have changed over time and frequently overlap. Indeed, in some sections of the manuscript Authors mentioned different types of HDP. I recommend Authors to modify the manuscript referring to HDP instead of gestational hypertension throughout all the text (including the title).

We have rewritten the introduction as suggested.

- Several important and widely available biomarkers studied in the first trimester for the prediction of HDP are missing (e.g. endothelin-1 and other markers of endothelial dysfunction doi: 10.1016/s0020-7292(99)00180-0., arterial stiffness doi: 10.1097/HJH.0000000000001664.). One of the most important ignored biomarker is serum uric acid and the recently evaluated serum uric acid/Creatinine ratio. Uric acid and uric acid normalized for kidney function have shown to hold prognostic value in HDP (doi: 10.1038/s41440-023-01181-6; doi: 10.1097/HJH.0000000000003472) being also included in the recommended biochemical analytes to be measured in women at high-risk (see ESC guidelines and ISSHP guidelines). 

We have addressed the suggested biomarkers with the exception of the arterial stifness because the work deals with biochemical markers, and measurements of vascular stiffness are based on biophysical methods.

- Authors stated that PAPP-A and PlGF are the most important biochemical markers for the first trimester prediction of HDP development. However, these well studied markers have shown in multiple high-powered studies to have poor predictive values (e.g. doi: 10.1002/pd.5001; doi: 10.1186/s12884-015-0608-y). Authors should focus on novel biomarkers and current literature on less-studied biomarkers in order to provide evidence for future studies to develop a multiparametric risk score that can ACCURATELY predict the risk for HDP development. This is of crucial importance since, as Authors stated, the only therapeutic measure is prevention and HDP incidence is dramatically increasing worldwide.

We did not deliberately focus on all the markers that have ever been studied, but mainly on those that have a clinical relevance for predictive models and the real work of doctors on a daily basis. We know that they are not perfect, because as the studies show, the data are contradictory, but for today they are available for practical use.

Best wishes,

Magda Rybak-Krzyszkowska

Reviewer 2 Report

For first.

The theme is very interesting and demanding. Although threre are many researches connected to hyperthension in pregnancy, there is still high morbidity of newborns and mothers due tu HTA in pregnancy and preeeclampsia.

Therefore, it is good to have new review article to see if there are some novelties in researches and wich way to proceed.

But, there are some issues that are worying me...

Major issue is that in the title the autors are talking about early biomarkers of gestational hyperthension. O the other hand, results show review based on 15 researches, and half of them are talking about small for gestational age and fetal growth restriction of newborns and conection of biomarkers with those morbidities. There is association of gestational hyperthension and FGR but..Or change the title and basis for research, or put greater point on gestational hyperthension.

Also, there is a difference between gestational hyperthension and preeclampsia, as authors know. So, maybe to change the title? Reconsider..

Now , minor changes

Abstract

1. Lines 25, 26 and 27: please write full names of PAPP -A,PIGF, sFLT, AFP and beta HCG when you mention them for first time.

Introduction

Line 47...maybe it would be better to say elevated risk for fetal asphyxia

Lines 55 and 56...Hypertension may affect the integrity of wndotelium...in what part of body? Explain it more.

Line 57...complications as intravascular coagulation and histopatological lesions...Unclear...explain it better

Lines 62....it might be better to say newborns then babies.

Lines 76 and 77..Administration of Aspirin goes from 12 th untill 35th week of gestation. After that it is not reccomended because of its influence on Ductus Botali closure of fetus. It reduces gestational hyperthension in hihgh risk women primarly, not preterm birth. Change the sentence.

Also reconstruct sentences inlines 78-80, it is unclear.

 Line 81.I guess you ment PAPP-A , not PAPPA.

Also, its the risk higher of 1:100 , not risk factors above that value. Also, explain these values in other countries...If there is other value of risk you have to name it and explain it.

Lines 84-87...reshape it, its unclear.

Line 91...What compounds. Explain a bit more.

Lines 99, 100..all shorts write with full names also, its their first mention in text.

Line 102...what other complications? Explain a bit

Line 104...what mechanisms? Name them at least

Lines 111-116..change the aim...It should be to review the literature and see what  potential biomarkers for HTA in pregnancy have been found untill now in literture.

Methods

Line 170...put reference as numeber, not surname and year

In results, as i have allready said,  i have issue. Or you will talk about researches about biomarkers for gestational yperthension only, or you will change review title and methods, and include biomarkers for SGA newborns also

Introduction part has some parts that looks like inadequate traslation on english. Introduction part has to be edited apropriatelly.

Author Response

Dear Reviewer,

Thank you for your valuable comments. we have adjusted our work and made the suggested changes. below we respond point by point to the comments.

For first.

The theme is very interesting and demanding. Although threre are many researches connected to hyperthension in pregnancy, there is still high morbidity of newborns and mothers due tu HTA in pregnancy and preeeclampsia.

Therefore, it is good to have new review article to see if there are some novelties in researches and wich way to proceed.

But, there are some issues that are worying me...

Major issue is that in the title the autors are talking about early biomarkers of gestational hyperthension. O the other hand, results show review based on 15 researches, and half of them are talking about small for gestational age and fetal growth restriction of newborns and conection of biomarkers with those morbidities. There is association of gestational hyperthension and FGR but..Or change the title and basis for research, or put greater point on gestational hyperthension.

Also, there is a difference between gestational hyperthension and preeclampsia, as authors know. So, maybe to change the title? Reconsider..

Thank you for your comment and suggestions, we have changed the title to:  Role of first trimester screening biochemical markers to predict hypertensive pregnancy disorders and SGA neonates – a narra-tive review

Now , minor changes

Abstract

  1. Lines 25, 26 and 27: please write full names of PAPP -A,PIGF, sFLT, AFP and beta HCG when you mention them for first time.

 Corrected as suggested.

Introduction

Line 47...maybe it would be better to say elevated risk for fetal asphyxia

 Corrected as suggested.

Lines 55 and 56...Hypertension may affect the integrity of wndotelium...in what part of body? Explain it more.

We have explained as suggested.

Line 57...complications as intravascular coagulation and histopatological lesions...Unclear...explain it better

 Corrected as suggested.

Lines 62....it might be better to say newborns then babies.

 Corrected as suggested.

Lines 76 and 77..Administration of Aspirin goes from 12 th untill 35th week of gestation. After that it is not reccomended because of its influence on Ductus Botali closure of fetus. It reduces gestational hyperthension in hihgh risk women primarly, not preterm birth. Change the sentence.

The sentense has been changed.

Also reconstruct sentences inlines 78-80, it is unclear.

 Corrected as suggested.

 Line 81.I guess you ment PAPP-A , not PAPPA.

 Corrected as suggested.

Also, its the risk higher of 1:100 , not risk factors above that value. Also, explain these values in other countries...If there is other value of risk you have to name it and explain it.

We have added recommendations form different regions of wolrd.

Lines 84-87...reshape it, its unclear.

 Corrected as suggested.

Line 91...What compounds. Explain a bit more.

We have explained as suggested.

Lines 99, 100..all shorts write with full names also, its their first mention in text.

 Corrected as suggested.

Line 102...what other complications? Explain a bit

We have explained as suggested.

Line 104...what mechanisms? Name them at least        

We have explained as suggested.

Lines 111-116..change the aim...It should be to review the literature and see what  potential biomarkers for HTA in pregnancy have been found untill now in literture.

Rewritten as suggested.

Methods

Line 170...put reference as numeber, not surname and year

 Corrected as suggested.

In results, as i have allready said,  i have issue. Or you will talk about researches about biomarkers for gestational yperthension only, or you will change review title and methods, and include biomarkers for SGA newborns also

We have changed the title.

Best wishes

Magda Rybak-Krzyszkowska

Reviewer 3 Report

Well-written review article. References neatly follow the text. Methodologically written, interesting for a wider readership

Early screening for gestational hypertension with the use of the biochemical markers for preeclampsia prediction – a narrative  review is well written in terms of explaining preeclampsia during pregnancy, outcomes, and biochemical markers. the authors set the methodology as a cross-sectional study that includes current knowledge and a very interesting and unexplored field of perinatology, also comparing the current cases in which SGA or LGA occurs in newborns. An excellently laid out and written paper, with references accompanying the introduction and discussion.

Author Response

Dear Reviewer,
Thank you for reading our article and reviewing it. Thank you for your favorable comments on our manuscript. 

Best wishes,

Magda Rybak-Krzyszkowska

Round 2

Reviewer 1 Report

The Authors correctly addressed my major concerns.

Reviewer 2 Report

This text is now more unedrstandable AND clear. i AGREE TO PUBLISH IT.